# SpecFormer: Guarding Vision Transformer Robustness via Maximum Singular Value Penalization

## Abstract

Vision Transformers (ViTs) have gained prominence as a preferred choice for a wide range of computer vision tasks due to their exceptional performance. However, their widespread adoption has raised concerns about security in the face of malicious attacks. Most existing methods rely on empirical adjustments during the training process, lacking a clear theoretical foundations. In this study, we address this gap by introducing **SpecFormer**, specifically designed to enhance ViTs' resilience against adversarial attacks, with support from carefully derived theoretical guarantees. We establish local Lipschitz bounds for the self-attention layer and introduce a novel approach, **Maximum Singular Value Penalization (MSVP)**, to attain precise control over these bounds. We seamlessly integrating MSVP into ViTs' attention layers, using the power iteration method for enhanced computational efficiency. The modified model, SpecFormer, effectively reduces the spectral norms of attention weight matrices, thereby enhancing network local Lipschitzness. This, in turn, leads to improved training efficiency and robustness. Extensive experiments on CIFAR and ImageNet datasets confirm SpecFormer's superior performance in defending against adversarial attacks.

## 1 Introduction

Vision Transformer (ViT) (Dosovitskiy et al., 2020) has gained increasing popularity in computer vision. Owing to its superior performance, ViT has been widely applied to image classification (Xu et al., 2022), object detection (Carion et al., 2020), semantic segmentation (Strudel et al., 2021), and video understanding (Arnab et al., 2021). More recently, the popular vision foundation models (Radford et al., 2021; Dehghani et al., 2023) also adopt ViT as the basic module. Unlike CNNs (Simonyan & Zisserman, 2014), a classical vision backbone, ViT initially divides images into non-overlapping patches and leverages the self-attention mechanism (Vaswani et al., 2017) for feature extraction.

Despite its popularity, security concerns related to ViT have recently surfaced as a critical issue. Studies have demonstrated that ViT is vulnerable to malicious attacks (Fu et al., 2022; Lovisotto et al., 2022), which compromises its performance and system security. Adversarial examples, which are created by adding trainable perturbations to the original inputs to produce incorrect outputs, are one of the major threats in machine learning security. Different attacks, such as FGSM (Goodfellow et al., 2015), PGD (Madry et al., 2018), and CW attack (Carlini & Wagner, 2017), have significantly impeded neural networks including both CNNs (Simonyan & Zisserman, 2014; Szegedy et al., 2015; He et al., 2016) and ViT (Dosovitskiy et al., 2020; Touvron et al., 2021a; d'Ascoli et al., 2021).

This work aims at improving the adversarial robustness of ViT. Prior arts (Bhojanapalli et al., 2021; Naseer et al., 2021; Bai et al., 2021b) have empirically investigated the intrinsic robustness of ViT compared with CNNs. Bhojanapalli et al. (2021) found out that when pre-trained with a sufficient amount of data, ViTs are at least as robust as the CNNs (Simonyan & Zisserman, 2014) on a broad range of perturbations. Bai et al. (2021b) stated that CNNs can be as robust as ViT against adversarial attacks if they properly adopt Transformers' training recipes. Furthermore, Shao et al. (2021); Paul & Chen (2022) used frequency filters to discover that the success of ViT' robustness lies in its lower sensitivity to high-frequency perturbations. But later works (Fu et al., 2022; Lovisotto et al., 2022;

Wang et al., 2022b) suggest that adding an attention-aware loss to ViTs can manipulate its output, resulting in lower robustness than CNNs.

Other than the robustness analysis of ViT, there are some empirical and theoretical efforts in designing algorithms to enhance its robustness. Empirically, Debenedetti et al. (2022) proposed a modified adversarial training recipe of more robust ViT by omitting the heavy data augmentations used in standard training. Mo et al. (2022) found that masking gradients from attention blocks or masking perturbations on some patches during adversarial training can greatly improve the robustness of ViT. While prior arts have mainly focused on improvements from an empirical perspective, they do not pay attention to the underlying theoretical principles governing self-attention and model robustness.

Some recent research (Zhou et al., 2022; Wang et al., 2022a; Qi et al., 2023) have investigated the theoretical properties of the self-attention mechanisms in ViTs. However, they focused on studying the stability during training (Wang et al., 2022a; Takase et al., 2022) without establishing an explicit link to robustness. While some scattered understandings about the robustness of attention mechanisms do exist, their applicability and reliability remain unclear. Therefore, it is imperative to conduct a comprehensive exploration on the robustness of ViT from *both* the theoretical and empirical perspectives. By doing so, we can possess a more thorough understanding of what influences the robustness of ViT, which is essential for the continued development of powerful and reliable deep learning models, especially large foundation models.

In this work, we propose SpecFormer, a simple yet effective approach to enhance the adversarial robustness of ViT. Concretely, we first provide a rigorous theoretical analysis of model robustness from the perspective of Lipschitz continuity (Murdock, 1999). Our analysis shows that we can easily control the Lipschitz continuity of self-attention by adding additional penalization. SpecFormer seamlessly integrate our proposed Maximum Singular Value Penaliztaion (MSVP) algorithm into each attention layer to help improve model stability. We further adopt the power iteration algorithm (Burden et al., 2015) to accelerate optimization. Our approach is evaluated on four public datasets, namely, CIFAR-10/100 (Krizhevsky et al., 2009), ImageNet (Deng et al., 2009) and Imagenette (Howard, 2019) across different ViT variants. Our SpecFormer achieves superior performance in *both* clean and robust accuracy. Under standard training, our approach improves robust accuracy by $2.79\%$ and $2.89\%$ against FGSM (Goodfellow et al., 2015) and PGD (Madry et al., 2018) attacks, respectively; under adversarial training, SpecFormer outperforms the best counterparts by $3.35\%$ and $2.58\%$ on CW (Carlini & Wagner, 2017) and PGD (Madry et al., 2018) attacks. The clean accuracy is also improved by $1.67\%$, and $3\%$ on average in both settings.

Our main contributions can be summarized as follows:

- We provide a comprehensive theoretical analysis of the Lipschitz continuity of Vision Transformer robustness and compare it with existing bounds in the literature.
- We propose **SpecFormer**, a robust Transformer architecture that applies **Maximum Spectral Value Penalization (MSVP)** to each attention layer to enhance adversarial robustness.
- Extensive experiments demonstrate the superiority of our approach in both clean and robust accuracy against different adversarial attacks.

## 2 RELATED WORK

### 2.1 ADVERSARIAL ROBUSTNESS

The early work of adversarial robustness (Szegedy et al., 2013) discovered that although deep networks are highly expressive, their learned input-output mappings are fairly discontinuous to some extent. Therefore, people can apply an imperceptible perturbation that maximizes the network's prediction error to cause misclassification of images. Since then, a vast amount of research (Goodfellow et al., 2015; Nguyen et al., 2015; Papernot et al., 2016a) has been done in adversarial attack (Carlini & Wagner, 2017; Chen et al., 2018), defense (Papernot et al., 2016b; Xie et al., 2018), robustness (Zheng et al., 2016), and theoretical understanding (Pang et al., 2022; Zhang et al., 2019a).

The problem of overall adversarial robustness can be formulated as a min-max optimization, $\min_{\theta} \max_{\delta} \mathcal{L}(f(\mathbf{x} + \boldsymbol{\delta}), y)$. The solution to the inner maximization problem w.r.t the input perturbations $\boldsymbol{\delta}$ corresponds to generating adversarial samples (Goodfellow et al., 2015; Madry et al., 2018),

while the solution to the outer minimization problem w.r.t the model parameters $\boldsymbol{\theta}$ corresponds to adversarial training (Tramèr et al., 2018; Bai et al., 2021a), which is an irreplaceable method for improving adversarial robustness and plays a crucial role in defending against adversarial attacks.

Existing attack approaches focus on deriving more and more challenging perturbations $\boldsymbol{\delta}$ to explore the limits of neural network adversarial robustness. For instance, Fast Gradient Sign Method (FGSM) (Goodfellow et al., 2015) directly take the sign of the derivative of the training loss w.r.t the input perturbation as $\boldsymbol{\delta} \leftarrow \boldsymbol{\delta} + \alpha \operatorname{sign} \left( \nabla_{\boldsymbol{\delta}} \mathcal{L}(f_{\boldsymbol{\theta}}(\mathbf{x} + \boldsymbol{\delta}), y) \right)$. Projected Gradient Descent (PGD) method (Madry et al., 2018) updates the perturbations by taking the sign of the loss function derivative w.r.t the perturbation, projecting the updated perturbations onto the admissible space, and repeating multiple times to generate more powerful attacks: $\boldsymbol{\delta} \leftarrow \Pi_{\epsilon} \left( \boldsymbol{\delta} + \alpha \cdot \operatorname{sign} \left( \nabla_{\boldsymbol{\delta}} \mathcal{L} \left( f_{\boldsymbol{\theta}}(\mathbf{x} + \boldsymbol{\delta}), y \right) \right) \right)$.

## 2.2 ROBUSTNESS OF VISION TRANSFORMER

Regarding the robustness of Vision Transformers, the first question people are curious about is how they compare to the classic CNN structure in terms of robustness. To address this question, numerous papers (Szegedy et al., 2013; Shao et al., 2021; Bhojanapalli et al., 2021; Bai et al., 2021a) have conducted thorough evaluations of both visual model structures, analyzed possible causes for differences in robustness, and proposed solutions to mitigate the observed gaps. Building upon these understandings, Zhou et al. (2022) developed fully attentional networks (FANs) to enhance the robustness of self-attention mechanisms. They evaluate the efficacy of these models in terms of corruption robustness for semantic segmentation and object detection, achieving state-of-the-art results. Additionally, Debenedetti et al. (2022) compared the DeiT (Touvron et al., 2021a), CaiT (Touvron et al., 2021b), and XCiT (Ali et al., 2021) ViT variants in adversarial training and discovered that XCiT was the most effective. This discovery sheds light on the idea that Cross-Covariance Attention could be another viable option for improving the adversarial robustness of ViTs.

In terms of the theoretical understanding of the relationship between Transformers and robustness, Zhou et al. (2022) attribute the emergence of robustness in Vision Transformers to the connection of self-attention mechanisms with information bottleneck theory. This suggests that the stacking of attention layers can be broadly regarded as an iterative repeat of solving an information-theoretic optimization problem, which promotes grouping and noise filtering. Dasoulas et al. (2021) analyzed that the norm of the derivative of attention models is directly related to the uniformity of the softmax probabilities. If all attention heads have uniform probabilities, the norm will reach its minimum; if the whole mass of the probabilities is on one element, the norm will reach its maximum. Considering a linear approximation of the attention mechanism, the smaller the derivative norm, the tighter the changes when perturbations are introduced, thus enhancing the robustness of the Transformer. These findings have been supported by another paper (Lovisotto et al., 2022) that proposes an attention-aware loss to deceive the predictions of Vision Transformers. Their attack strategy exactly misguides the attention of all queries towards a single key token under the control of an adversarial patch, corresponding to the maximum derivative norm case described above.

## 3 PRELIMINARIES

Denote a clean dataset as $\mathcal{D} = \{\mathbf{x}_i, y_i\}_{i=1}^{N}$, where $\mathbf{x}$ and $y$ are the input and ground-truth label respectively, and denote the loss function as $\mathcal{L}$. We aim to investigate the robustness of the Vision Transformer (ViT) under two different training paradigms: standard supervised training and adversarial training (AT) (Madry et al., 2018). In standard training, we aim to learn a classification function $f_{\boldsymbol{\theta}}$ parameterized by $\boldsymbol{\theta}$, where the optimal parameter is $\boldsymbol{\theta}^* = \arg\min_{\boldsymbol{\theta}} \mathbb{E}_{(\mathbf{x},y) \in \mathcal{D}} \mathcal{L}(f_{\boldsymbol{\theta}}(\mathbf{x}), y)$.

While standard training is shown to be less resilient to adversarial attacks, AT is a major and effective paradigm for adversarial robustness. AT aims to improve a model's ability to resist malicious attacks by solving a min-max optimization problem and generating adversarial examples by adding strategic perturbations $\boldsymbol{\delta}$ within a given budget $\delta_0$, which can fool the model predictions, i.e., $f_{\boldsymbol{\theta}}(\mathbf{x} + \boldsymbol{\delta}) \neq y$. The model parameters are updated to reduce the classification error on the perturbed examples, resulting in enhanced robustness against adversarial attacks:

$$\boldsymbol{\theta}^* = \arg\min_{\boldsymbol{\theta}} \mathbb{E}_{(\mathbf{x},y) \in \mathcal{D}} \max_{\|\boldsymbol{\delta}\|_2 \leq \delta_0} \mathcal{L}\left( f_{\boldsymbol{\theta}}(\mathbf{x} + \boldsymbol{\delta}), y \right), \tag{1}$$

where $\delta_0$ bounds the magnitude of $\boldsymbol{\delta}$ to prevent unintended semantic changes caused by perturbation.

This work aims to improve the adversarial robustness of ViT in *both* AT and non-AT scenarios. We show that by adding a slight penalization, its robustness can be greatly enhanced in both settings.

# 4 THEORETICAL ANALYSIS

Bounding the *global* Lipschitz constant of a neural network is a commonly used method to provide robustness guarantees (Cisse et al., 2017; Leino et al., 2021). However, the global Lipschitz bound can be loose because it needs to hold for all points in the input domain, including inputs that are far apart. This can greatly reduce clean accuracy in empirical comparisons (Huster et al., 2019; Madry et al., 2018). On the other hand, a local Lipschitz constant bounds the norm of output perturbations only for inputs within a small region, typically selected as a neighborhood around each data point. This aligns perfectly with the scenario of adversarial robustness, as discussed in Section 2.1, where perturbations attempt to affect the model's output within a budget constraint. Local Lipschitz bounds are superior because they produce tighter bounds by considering the geometry in a local region, often leading to much better robustness (Hein & Andriushchenko, 2017; Zhang et al., 2019b).

The core of our study is to bridge the gap between local Lipschitz continuity and adversarial robustness in ViTs. Primary distinctions of ViT lie in the LayerNorm and self-attention layers. The seminal work (Kim et al., 2021) proved the dot-product is not globally Lipschitz continuous and the LayerNorm is Lipschitz continuous (see Kim et al. (2021), Appendix N). Therefore, we only need to modify that concept and prove that the dot-product self-attention is local Lipschitz continuous. By utilizing the local Lipschitz continuity, the adversarial robustness can be strengthened.

**Definition 4.1** (Local Lipschitz Continuity). *Suppose $\mathcal{X} \subseteq \mathbb{R}^d$ is open. A function, denoted as $f$, is considered locally Lipschitz continuous with respect to the p-norm, denoted as $\| \cdot \|_p$, if, for any given point $\mathbf{x}_0$, there exists a positive constant $C$ and a positive value $\delta_0$ such that whenever $\|\mathbf{x} - \mathbf{x}_0\|_p < \delta_0$, the following condition holds:*

$$\|f(\mathbf{x}) - f(\mathbf{x}_0)\|_p \le C \|\mathbf{x} - \mathbf{x}_0\|_p. \tag{2}$$

The smallest value of $C$ that satisfies the condition is called the local Lipschitz constant of $f$. From Eq. (2), we observe that a classifier exhibiting local Lipschitz continuity with a small $C$ experiences less impact on output predictions when subjected to budget-constrained perturbations. In this study, we harness the concept of local Lipschitz continuity to safeguard ViT against malicious attacks. Our primary focus is on ensuring that the attention layer maintains Lipschitz continuity in the vicinity of each input, and we employ optimization objectives to strengthen this property. This strategic approach enables us to bolster the output stability of ViT when facing adversarial attacks. We provide the formal definition of the local Lipschitz constant and the method for its calculation below.

**Definition 4.2** (Local Lipschitz Constant). *The $p$-local Lipschitz constant of a network $f(\mathbf{x})$ over an open set $\mathcal{X} \subseteq \mathbb{R}^d$ is defined as:*

$$\text{Lip}_p(f, \mathcal{X}) = \sup_{\substack{\mathbf{x}_1, \mathbf{x}_2 \in \mathcal{X} \\ \mathbf{x}_1 \ne \mathbf{x}_2}} \frac{\|f(\mathbf{x}_1) - f(\mathbf{x}_2)\|_p}{\|\mathbf{x}_1 - \mathbf{x}_2\|_p}. \tag{3}$$

If $f$ is smooth and $p$-local Lipschitz continuous over $\mathcal{X}$, the Lipschitz constant can be computed by upper bounding the norm of Jacobian.

**Theorem 4.1** (Calculation of Local Lipschitz Constant (Federer, 1969) ). *Let $f : \mathcal{X} \to \mathbb{R}^m$ be differentiable and locally Lipschitz continuous under a choice of p-norm $\| \cdot \|_p$. Let $\mathbf{J}_f(x)$ denote its total derivative (Jacobian) at $\mathbf{x}$. Then,*

$$\text{Lip}_p(f, \mathcal{X}) = \sup_{\mathbf{x} \in \mathcal{X}} \|\mathbf{J}_f(\mathbf{x})\|_p, \tag{4}$$

*where $\|\mathbf{J}_f(\mathbf{x})\|_p$ is the induced operator norm on $\mathbf{J}_f(\mathbf{x})$.*

The *global* Lipschitz constant, which takes into account the supremum over $\mathcal{X} = \mathbb{R}^d$, must ensure Eq. (2) even for distant $\mathbf{x}$ abd $\mathbf{x}_0$, which can render it imprecise and lacking significance when examining the local behavior of a network around a single input. We concentrate on 2-local Lipschitz constants, where $\mathcal{X} = B_2(\mathbf{x}_0, \delta_0) := \{\mathbf{x} : \|\mathbf{x} - \mathbf{x}_0\|_2 \le \delta_0\}$ represents a small $\ell_2$-ball with a radius

of $\delta_0$ centered around $\mathbf{x}_0$. The choice of the 2-norm for our investigation is motivated by two key reasons. First, the 2-norm is the most commonly used norm in Euclidean space. Second, and delving further into the rationale, as revealed in (Yoshida & Miyato, 2017), the sensitivity of a model to input perturbations is intricately connected to the 2-norm of the weight matrices, which in turn is closely linked to the principal direction of variation and the maximum scale change.

**Proposition 4.2** (Connection of Model Sensitivity and Maximum Singular Value (Yoshida & Miyato, 2017))**.** *In a small neighbourhood of* $\mathbf{x}_0$*, we can regard* $f_{\boldsymbol{\theta}}$ *as a linear function:* $\mathbf{x} \mapsto \mathbf{W}_{\boldsymbol{\theta},\mathbf{x}_0}\mathbf{x} + \boldsymbol{b}_{\boldsymbol{\theta},\mathbf{x}_0}$*, whose weights and biases depend on* $\boldsymbol{\theta}$ *and* $\mathbf{x}_0$*. For a small perturbation* $\delta$*, we have*

$$\frac{\|f_{\boldsymbol{\theta}}(\mathbf{x}_0+\boldsymbol{\delta})-f(\mathbf{x}_0)\|_2}{\|\boldsymbol{\delta}\|_2} = \frac{\|(\mathbf{W}_{\boldsymbol{\theta},\mathbf{x}_0}(\mathbf{x}+\boldsymbol{\delta})+\boldsymbol{b}_{\boldsymbol{\theta},\mathbf{x}_0})-(\mathbf{W}_{\boldsymbol{\theta},\mathbf{x}_0}\mathbf{x}+\boldsymbol{b}_{\boldsymbol{\theta},\mathbf{x}_0})\|_2}{\|\boldsymbol{\delta}\|_2} = \frac{\|\mathbf{W}_{\boldsymbol{\theta},\mathbf{x}_0}\boldsymbol{\delta}\|_2}{\|\boldsymbol{\delta}\|_2} \le \sigma_{max}\left(\mathbf{W}_{\boldsymbol{\theta},\mathbf{x}_0}\right). \quad (5)$$

The proposition above suggests that when the maximum singular value of the weight matrices is small, the function $f_{\boldsymbol{\theta}}$ becomes less sensitive to input perturbations, which can significantly enhance adversarial robustness. Inspired by this, we can re-examine the self-attention mechanism as a product of linear mappings and apply the same spirit to enhance the robustness of Vision Transformers.

**Re-examining Self-Attention as a Product of Linear Mapping Operations** We propose to reconsider self-attention, the fundamental module in ViT as a multiplication of three *linear mappings*, on which the aforementioned proposition on model robustness can be applied. Given an input $\mathbf{X} \in \mathbb{R}^{N \times d}$, the self-attention module in ViT is typically represented as:

$$\text{Attn}\left(\mathbf{X}, \mathbf{W}^Q, \mathbf{W}^K, \mathbf{W}^V\right) = \text{softmax}\left(\frac{\mathbf{X}\mathbf{W}^Q\left(\mathbf{X}\mathbf{W}^K\right)^\top}{\sqrt{D}}\right)\mathbf{X}\mathbf{W}^V, \quad (6)$$

where $\mathbf{W}^Q, \mathbf{W}^K, \mathbf{W}^V \in \mathbb{R}^{d \times D}$ are projection weight matrices corresponding to query, key, and value. $N, d$, and $D$ stand for the number of tokens, data dimension, and the hidden dimension of self-attention, respectively. By virtue of the intrinsic nature of self-attention, we conceptualized it as a three-way linear transformation, where the matrices are multiplied together:

$$\text{Attn}\left(\mathbf{X}, \mathbf{W}^Q, \mathbf{W}^K, \mathbf{W}^V\right) = \text{softmax}\left(\frac{\mathbf{X}\mathbf{W}^Q\left(\mathbf{X}\mathbf{W}^K\right)^\top}{\sqrt{D}}\right)\mathbf{X}\mathbf{W}^V = \text{softmax}\left(\frac{h_1(\mathbf{X})h_2(\mathbf{X})^\top}{\sqrt{D}}\right)h_3(\mathbf{X}),$$
$$(7)$$

where these linear mapping operations are formulated as:

$$h_1(\mathbf{X}) = \mathbf{X}\mathbf{W}^Q, \ h_2(\mathbf{X}) = \mathbf{X}\mathbf{W}^K, \ h_3(\mathbf{X}) = \mathbf{X}\mathbf{W}^V. \quad (8)$$

After reinterpreting the self-attention mechanism as a product of three linear mappings, we are inspired by Prop. 4.2 to readily discern that we can independently control the maximum singular values of these three weight matrices to imbue the model with adversarial robustness.

**Theorem 4.3.** *The self-attention layer is local Lipschitz continuous in* $B_2\left(\mathbf{X}_0, \delta_0\right)$

$$\text{Lip}_{local}(\text{Att}, \mathbf{X}_0) \le N(N+1)\left(\|X_0\|_F + \delta_0\right)^2\left[\left\|\mathbf{W}^V\right\|_2\left\|\mathbf{W}^Q\right\|_2\left\|\mathbf{W}^{K,\top}\right\|_2 + \left\|\mathbf{W}^V\right\|_2\right], \quad (9)$$

*If we make a more stringent assumption that all inputs to the self-attention layer are bounded, i.e.,* $\mathcal{X} \in \mathbb{R}^{N \times d}$ *represents a bounded open set, then we can determine the maximum of the input* $\mathbf{X}$ *as* $B = \max_{\mathbf{X} \in \mathcal{X}} \|\mathbf{X}\|_F$*. This allows us to establish a significantly stronger conclusion:*

$$\text{Lip}_{local}(\text{Att}, \mathbf{X}_0) \le N(N+1)\left(B + \delta_0\right)^2\left[\left\|\mathbf{W}^V\right\|_2\left\|\mathbf{W}^Q\right\|_2\left\|\mathbf{W}^{K,\top}\right\|_2 + \left\|\mathbf{W}^V\right\|_2\right]. \quad (10)$$

This framework provides us with theoretical support, allowing us to manage the local Lipschitz constant of the attention layer by controlling the maximum singular values of $\mathbf{W}^Q, \mathbf{W}^K, \mathbf{W}^V$.

## 5 SPECFORMER

Inspired by the above analysis, we propose **SpecFormer**, a more robust ViT against adversarial attacks, as illustrated in Figure 1. SpecFormer employs Maximum Singular Value Penalization (MSVP) with an approximation algorithm named power iteration to reduce the computational costs.

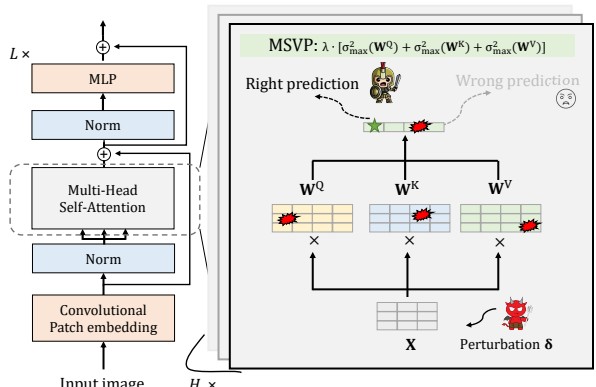

FIGURE 1: SpecFormer with MSVP.

**Algorithm 1** SpecFormer

1: ▷ Initialization:
2: **for** layer $\ell = 1$ to $L$ **do**
3:      Init $\boldsymbol{u}_{q/k/v}^\ell, \boldsymbol{v}_{q/k/v}^\ell$
4: **end for**
5: ▷ Forward:
6: A minibatch $\{(\mathbf{x}_1, y_1), \ldots, (\mathbf{x}_k, y_k)\}$
7: **for** layer $\ell = 1$ to $L$ **do**
8:      Pass minibatch through the layers
9:      **for** head $h = 1$ to $H$ **do**
10:          $\boldsymbol{v}^\ell \leftarrow (W^\ell)^\top \boldsymbol{u}^\ell, \boldsymbol{u}^\ell \leftarrow \mathbf{W}^\ell \boldsymbol{v}^\ell,$
11:          $\sigma^\ell \leftarrow \boldsymbol{u}^\ell \mathbf{W}^\ell \boldsymbol{v}^\ell,$
12:          Add $\lambda(\sigma^\ell)^2$ to $\mathcal{L}_{msvp}$.
13:      **end for**
14: **end for**
15: ▷ Backward:
16: Update $\theta$ by backprop: $\mathcal{L}_{cls} + \mathcal{L}_{msvp}$.

## 5.1 MAXIMUM SINGULAR VALUE PENALIZATION

MSVP aims to enhance the robustness of ViT by adding penalization to the self-attention layers. Concretely speaking, MSVP restricts the Lipschitz constant of self-attention layers by penalizing the maximum singular values of the linear transformation matrices $\mathbf{W}^Q, \mathbf{W}^K$, and $\mathbf{W}^V$. Denote the classification loss as $\mathcal{L}_{cls}$ such as cross-entropy, the overall training objective with MSVP is:

$$\mathcal{J} = \mathcal{L}_{cls} + \mathcal{L}_{msvp} = \mathcal{L}_{cls} + \lambda \cdot \left[\sigma_{\max}^2(\mathbf{W}^Q) + \sigma_{\max}^2(\mathbf{W}^K) + \sigma_{\max}^2(\mathbf{W}^V)\right], \quad (11)$$

where $\lambda$ is the trade-off hyperparameter. The original Transformer (Vaswani et al., 2017) adopts multi-head self-attention to jointly attend to information from different subspaces at different positions. In line with that spirit, MSVP can also be added in a multi-head manner. Incorporating the summation over all the heads and layers, the overall training objective with multi-head MSVP is:

$$\mathcal{J} = \mathcal{L}_{cls} + \mathcal{L}_{msvp} = \mathcal{L}_{cls} + \lambda \sum_{l=1}^{L} \sum_{h=1}^{H} \left[\sigma_{\max}^2(\mathbf{W}_l^{Q,h}) + \sigma_{\max}^2(\mathbf{W}_l^{K,h}) + \sigma_{\max}^2(\mathbf{W}_l^{V,h})\right], \quad (12)$$

where $\mathbf{W}_l^{*,h}$ denotes the $h^{th}$ head in the $l^{th}$ attention layer, and $*$ could be $Q, K$ or $V$. By constraining the maximum singular value of the weight matrices in the mapping, we can regulate the extent of output variations when subjected to attacks, thereby enhancing the model's robustness. The computation of maximum singular values can be seamlessly integrated into the forward process.

In the next section, we show how to perform efficient computation of maximum singular values using the theoretically guaranteed (Thm. 5.1) power iteration algorithm.

## 5.2 POWER ITERATION

Singular value decomposition (SVD) is the most direct way to calculate the maximum singular value. For any real matrix $\mathbf{A} \in \mathbb{R}^{m \times n}$, there exists a singular value decomposition of the form $\mathbf{A} = \mathbf{U}\boldsymbol{\Sigma}\mathbf{V}^\top$, where $\mathbf{U}$ is an $m \times m$ orthogonal matrix, $\boldsymbol{\Sigma}$ is an $m \times n$ diagonal matrix and $\mathbf{V}$ is an $n \times n$ orthogonal matrix. However, directly adopting SVD in MSVP will result in an additional time complexity of $\mathcal{O}(m^2 n + n^3)$, which poses a disastrous computational burden, particularly given the high hidden dimension of ViT and the vast number of attention layers used in the model.

In this section, we propose to adopt the power iteration algorithm as an alternative approach to efficiently calculate the maximum singular values for the $\mathbf{W}^Q, \mathbf{W}^K, \mathbf{W}^V$ weight matrices. The power iteration method (Burden et al., 2015) is a commonly used approach to approximate the maximum singular value with each iteration taking $\mathcal{O}(mn)$ time. By selecting an initial approximation vector $\boldsymbol{u}$ and $\boldsymbol{v}$, and then performing left and right matrix multiplication with the target matrix, we can obtain a reliable and accurate underestimate of the maximum singular value in a constant number of iterations. Moreover, as the algorithm is iterative in nature, we can incorporate the updates into the model's update process, permitting us to efficiently estimate the maximum singular values with

minimal cost. The following theorem provides a convergence guarantee of power iteration and the proof is in Appendix C. The complete training pipeline of SpecFormer is shown in Alg. 1.

**Theorem 5.1** (Convergence guarantee of the power iteration method (Mises & Pollaczek-Geiringer, 1929)). *Assuming the dominant singular value $\sigma_{max}(\mathbf{A})$ is strictly greater than the subsequent singular values and that $\mathbf{u}_0$ is initially selected at random, then there is a probability of $1$ that $\mathbf{u}_0$ will have a non-zero component in the direction of the eigenvector linked with the dominant singular value. Consequently, the convergence will be geometric with a ratio of $\left| \frac{\sigma_2(\mathbf{A})}{\sigma_{max}(\mathbf{A})} \right|$.*

## 6 EXPERIMENTS

### 6.1 SETUP

**Datasets.** We adopt four popular benchmark datasets: CIFAR10,CIFAR100 (Krizhevsky et al., 2009), ImageNet (Deng et al., 2009), and Imagenette (Howard, 2019). These datasets are commonly adopted in studies involving the robustness of vision models (Pang et al., 2022; Mo et al., 2022). CIFAR-10 and CIFAR-100 datasets each comprise $60,000$ images, categorized into 10 classes and 100 classes, respectively. ImageNet encompasses over $1.2$ million training images and $50,000$ test images, distributed across $1,000$ classes. Imagenette is a subset consisting of 10 classes that are easy to classify, selected from the ImageNet. It is often employed as a suitable proxy (Mo et al., 2022; Wang et al., 2023) for evaluating the performance of models on the more extensive ImageNet.

**Baselines.** We compare SpecFormer with several notable baselines. Specifically, we evaluate Spec-Former against LipsFormer (Qi et al., 2023), which introduces Lipschitz continuity into ViT through a novel scaled cosine similarity attention (SCSA) mechanism and by replacing other unstable Transformer components with Lipschitz continuous equivalents. Additionally, we incorporate two other baseline models that implement Lipschitz continuity through different mechanisms: L2 multi-head attention (Kim et al., 2021) and pre-softmax Lipschitz normalization (Dasoulas et al., 2021). However, it is worth noting that the L2 multi-head attention approach requires $\mathbf{W}^{Q,h} = \mathbf{W}^{K,h}$, which is overly restrictive, potentially compromising the model's representation power. We denote the L2 attention and Lipschitz normalization methods as L2Former and LNFormer, respectively.

**Implementation details.** In line with prior work (Mo et al., 2022), our evaluation spans across three distinct ViT backbones: the vanilla ViT (Dosovitskiy et al., 2020), DeiT (Touvron et al., 2021a), and ConViT (d'Ascoli et al., 2021). This approach enables us to perform a thorough validation and assess the general effectiveness of the MSVP algorithm. To ensure a fair comparison, we adopt the training strategy outlined in (Mo et al., 2022). Specifically, we use the SGD optimizer with a weight decay of $0.0001$ and a learning rate of $0.1$ for $40$ epochs. By default, we apply both CutMix and Mixup data augmentation. We employ FGSM (Goodfellow et al., 2015) and PGD-2 (Madry et al., 2018) attacks for standard training, with an attack radius of $2/255$. For adversarial training, we use CW-20 (Carlini & Wagner, 2017) and PGD-20 (Madry et al., 2018) attacks, with an attack radius of $8/255$. In the case of the baseline models (Lipsformer, L2former, and LNFormer), we strictly adhere to the training protocols described in their respective papers.

### 6.2 MAIN RESULTS

**SpecFormer effectively controls the Lipschitz continuity of Transformers with minor modifications.** From Table 1, 2 and 6, we can see that our proposed SpecFormer achieves the best results among all the competitors. Specifically, on standard training, our SpecFormer improves over the state-of-the-art robust ViT approaches by **2.79%** and **2.89%** in terms of robust accuracy on FGSM and PGD attacks, respectively, on average. Besides, we also improve the standard accuracy by **1.67%**. Using adversarial training, our SpecFormer outperforms the best counterparts by **3.35%** and **2.58%** on CW and PGD attacks, respectively. Our method improves the clean accuracy of the ViT model by **3.80%**. It significantly enhances the model's robustness and its original performance.

We have more observations from the results. 1) Some baseline methods exhibit inferior performance when compared to vanilla ViTs, potentially attributable to the imposition of excessively stringent Lipschitz continuity constraints, which limit the model's expressive capacity. 2) In contrast to other methods that ensure Lipschitz continuity in Transformers through self-attention mechanism modifications, our SpecFormer achieves this by adding a straightforward penalty term to the attention

TABLE 1: Performance (%) of SpecFormer with different ViT variants on benchmark datasets under *standard* training (using ImageNet-1k pre-trained weights). The best results are in **bold**.

| Model | Method | CIFAR-10 | | | CIFAR-100 | | | Imagenette | | |
|-------|--------|----------|------|-------|-----------|------|-------|------------|------|-------|
| | | Standard | FGSM | PGD-2 | Standard | FGSM | PGD-2 | Standard | FGSM | PGD-2 |
| ViT-S | LipsFormer (Qi et al., 2023) | 71.13 | 31.48 | 4.17 | 40.05 | 9.92 | 1.36 | 86.80 | 36.60 | 30.20 |
| | L2Former (Kim et al., 2021) | 79.65 | 39.98 | 13.39 | 53.20 | 15.35 | 5.92 | 92.80 | 58.00 | 37.80 |
| | LNFormer (Dasoulas et al., 2021) | 75.82 | 33.72 | 7.75 | 48.81 | 13.04 | 7.27 | 92.00 | 49.00 | 41.80 |
| | ViT (Dosovitskiy et al., 2020) | 87.09 | 45.56 | 22.35 | 63.52 | 19.82 | 7.01 | 94.20 | 72.60 | 48.00 |
| | SpecFormer (Ours) | **88.52** | **50.58** | **29.53** | **69.78** | **23.92** | **9.67** | **97.20** | **84.20** | **61.60** |
| DeiT-Ti | LipsFormer (Qi et al., 2023) | 72.54 | 36.14 | 3.46 | 39.72 | 8.66 | 0.96 | 79.40 | 30.60 | 8.00 |
| | L2Former (Kim et al., 2021) | 78.09 | 36.64 | 5.05 | 49.02 | 12.63 | 2.56 | 82.40 | 51.00 | 6.00 |
| | LNFormer (Dasoulas et al., 2021) | 77.16 | 34.78 | 3.81 | 52.50 | 14.40 | **2.83** | 80.20 | 44.80 | 9.20 |
| | ViT (Dosovitskiy et al., 2020) | 86.40 | **46.10** | 14.46 | 62.79 | 19.89 | 2.35 | 90.00 | 64.20 | 13.00 |
| | SpecFormer (Ours) | **87.42** | 45.71 | **18.10** | **64.14** | **20.93** | 1.61 | **92.20** | **70.20** | **26.20** |
| ConViT-Ti | LipsFormer (Qi et al., 2023) | 79.71 | 38.47 | 7.09 | 48.08 | 10.84 | 1.57 | 90.60 | 44.40 | 24.60 |
| | L2Former (Kim et al., 2021) | 81.33 | 40.17 | 12.76 | 49.86 | 13.86 | 2.03 | 93.40 | 62.20 | 30.80 |
| | LNFormer (Dasoulas et al., 2021) | 75.48 | 28.67 | 3.56 | 51.13 | 11.62 | 2.41 | 84.20 | 36.20 | 15.00 |
| | ViT (Dosovitskiy et al., 2020) | **87.78** | **48.89** | 20.26 | 64.68 | **22.94** | **4.96** | **93.20** | 68.80 | **40.80** |
| | SpecFormer (Ours) | 87.49 | 47.64 | **20.53** | **65.57** | 21.78 | 4.10 | 92.60 | **69.00** | 28.60 |

TABLE 2: Performance (%) of SpecFormer with different ViT variants on benchmark datasets under *adversarial* training (using ImageNet-1k pre-trained weights). The best results are in **bold**.

| Model | Method | CIFAR-10 | | | CIFAR-100 | | | Imagenette | | |
|-------|--------|----------|-------|--------|-----------|-------|--------|------------|-------|--------|
| | | Standard | CW-20 | PGD-20 | Standard | CW-20 | PGD-20 | Standard | CW-20 | PGD-20 |
| ViT-S | LipsFormer (Qi et al., 2023) | 41.54 | 24.20 | 27.50 | 24.08 | 10.47 | 13.04 | 50.00 | 31.00 | 34.80 |
| | L2Former (Kim et al., 2021) | 63.22 | 33.55 | **35.78** | 37.20 | 13.69 | 15.65 | 84.80 | 54.60 | 54.60 |
| | LNFormer (Dasoulas et al., 2021) | 56.04 | 29.92 | 32.74 | 26.37 | 9.84 | 11.63 | 81.00 | 46.80 | 48.80 |
| | ViT (Dosovitskiy et al., 2020) | 71.76 | **34.34** | 35.49 | 36.45 | 11.97 | 12.89 | 89.60 | 62.80 | 62.40 |
| | SpecFormer (Ours) | **72.73** | 31.84 | 31.90 | **41.46** | **12.80** | **13.54** | **91.60** | **67.00** | **67.00** |
| DeiT-Ti | LipsFormer (Qi et al., 2023) | 39.72 | 24.05 | 26.77 | 23.09 | 9.82 | 12.03 | 39.00 | 27.60 | 28.40 |
| | L2Former (Kim et al., 2021) | 60.85 | 34.29 | 36.63 | 36.67 | 14.23 | 16.48 | 77.40 | 41.40 | 43.20 |
| | LNFormer (Dasoulas et al., 2021) | 54.32 | 29.85 | 32.96 | 28.65 | 11.36 | 13.94 | 72.20 | 39.00 | 42.20 |
| | ViT (Dosovitskiy et al., 2020) | 71.71 | 37.15 | 38.74 | 40.89 | 15.25 | 17.40 | 79.00 | 40.60 | 41.40 |
| | SpecFormer (Ours) | **80.03** | **48.52** | **51.10** | **44.48** | **16.36** | **18.21** | **82.20** | **46.20** | **46.00** |
| ConViT-Ti | LipsFormer (Qi et al., 2023) | 56.83 | 32.27 | 35.04 | 31.50 | 14.56 | 17.17 | 60.80 | 34.00 | 38.20 |
| | L2Former (Kim et al., 2021) | 39.36 | 23.84 | 26.42 | 16.53 | 8.06 | 9.65 | 10.00 | 10.00 | 10.00 |
| | LNFormer (Dasoulas et al., 2021) | 49.03 | 28.68 | 31.68 | 29.12 | 13.00 | 15.65 | 72.20 | 39.00 | 42.20 |
| | ViT (Dosovitskiy et al., 2020) | 53.09 | 30.87 | 33.63 | 31.54 | **14.65** | **17.24** | 68.00 | 41.40 | 43.60 |
| | SpecFormer (Ours) | **67.05** | **38.72** | **41.58** | **37.92** | 14.50 | 16.78 | **86.60** | **47.20** | **46.20** |

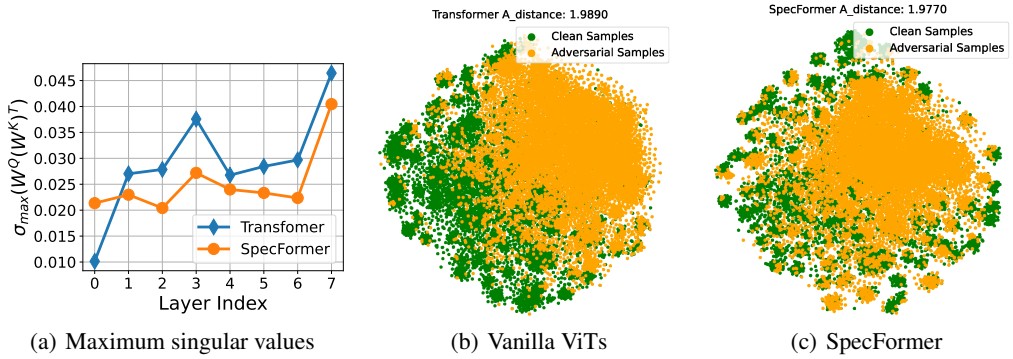

|     | (a) Maximum singular values | (b) Vanilla ViTs | (c) SpecFormer |

FIGURE 2: The analysis of MSVP. (a) Maximum singular value comparison between MSVP and the vanilla Transformer. (b)&(c) tSNE (Van der Maaten & Hinton, 2008) feature visualization.

layers, without altering the original self-attention mechanism. SpecFormer's simplicity and versatility allow seamless integration into various ViT architectures, preserving their flexibility. Experiment results on ImageNet and hyperparameter analysis is provided in Appendix E.1 and E.3.

## 6.3 ANALYZING THE EFFICACY OF MSVP

**Analyzing the maximum singular values.** We analyze the effectiveness of the proposed MSVP algorithm, which is the core of SpecFormer. Figure 2(a) shows the maximum singular values of both ViT and our SpecFormer. It indicates that with minimal additional cost, our SpecFormer effectively

limits the maximum singular values across all layers, which are smaller than those of vanilla ViTs. Hence, our proposed MSVP algorithm can effectively control the maximum singular value of the attention layers, thereby ensuring stability even under extreme perturbations.

**Feature visualization.** We further illustrate t-SNE feature visualizations of both vanilla ViT and our SpecFormer in Figures 2(b) and 2(c), employing the t-SNE technique as introduced by (Van der Maaten & Hinton, 2008). Adversarial samples introduce subtle perturbations aimed at diverting corrupted samples away from their correct categories. In these figures, each distinct cluster represents a class. While vanilla ViTs encounter difficulties in recovering adversarial samples that have been pushed into other categories, our modified model, SpecFormer, adeptly realigns the corrupted samples with their original true classes. This contrast serves to demonstrate the significant enhancement of ViTs' adversarial robustness by our method.

**Feature distribution distance.** To further support this contrast quantitatively, we compute the $\mathcal{A}$-distance (Ben-David et al., 2006), which quantifies the similarity between two distributions, with a lower value indicating a higher degree of similarity. As shown in Figure 2(b) and 2(c), Spec-Former achieves a lower $\mathcal{A}$-distance compared to vanilla ViTs, highlighting its capability to effectively counteract the distribution shift induced by adversarial perturbations. Consequently, our approach enhances the safety and robustness of ViTs. To sum up, these analyses demonstrate that our SpecFormer can both efficiently and effectively learn robust features w.r.t the adversarial attacks.

**Computation cost.** Finally, we analyze the computation cost of our proposed method. Table 3 displays the relative running time difference for each model to complete 10 training steps under both standard and adversarial conditions (with vanilla ViT set to 1). The table reveals that our proposed method exhibits the lowest additional computational costs compared to other baseline models, underscoring the computational efficiency of our approach.

TABLE 3: Relative running time difference for 10-steps of training (vanilla ViT=1). The lowest additional computation costs are in **bold**.

| Relative Diff | LipsFormer | L2Former | LNFormer | TransFormer | SpecFormer |
|---|---|---|---|---|---|
| Standard Training | +4.5 | +1.5 | +3.5 | 0.0 | +**0.5** |
| Adversarial Training | +11.3 | +3.3 | +10.7 | 0.0 | +**1.5** |

## 7 LIMITATIONS

In the realm of adversarial robustness, a challenging yet frequently discussed issue is the relationship between clean accuracy and robust accuracy. In this study, we are no exception, as our experiments have revealed the intricate balance between clean accuracy and robust accuracy through the adjustment of hyperparameters. Investigating and modeling this relationship further using theoretical approaches will be a primary focus for our future research endeavors. Furthermore, as mentioned earlier, the strict adherence to complete Lipschitz continuity in mathematical terms can prove overly stringent and potentially limit the diversity of neural network representations, consequently affecting performance. Therefore, exploring theoretical models that address the tradeoff between Lipschitz continuity and expressiveness is also a worthy avenue for inquiry. Last but not least, it is essential to note that our method has only undergone validation against a limited number of attack scenarios. Conducting additional assessments across a broader spectrum of attacks, including but not limited to AutoAttack (Croce & Hein, 2020), backdoor attacks (Li et al., 2022), and BIM (Feinman et al., 2017), is imperative to provide a more comprehensive evaluation.

## 8 CONCLUSION

This paper presents a theoretical analysis of the self-attention mechanism in ViTs and establishes a relationship between the adversarial robustness of ViTs and Lipschitz continuity theory. We demonstrate that the Lipschitz continuity of ViTs in the vicinity of the input can be enhanced by penalizing the maximum singular values of the attention weight matrices. Extensive experiments validate the effectiveness of our approach, showcasing superior performance in terms of both clean and robust accuracy against various adversarial attacks.

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

## A  GENERAL OPTIMIZATGION OBJECTIVE FOR MSVP

Here we provide the most general form of our proposed **Maximum Singular Value Penalization (MSVP)**. In this context, we employ three hyperparameters to flexibly adjust the strength of the maximum singular value penalties for different attention branches. This aspect is crucial for achieving improved performance and better adaptability within the self-attention mechanism, playing distinct and non-interchangeable roles for $\mathbf{W}^Q$, $\mathbf{W}^K$, and $\mathbf{W}^V$. Denote the classification loss as $\mathcal{L}_{cls}$ such as cross-entropy loss, the overall general training objective with MSVP is:

$$\mathcal{J} = \mathcal{L}_{cls} + \mathcal{L}_{msvp} = \mathcal{L}_{cls} + \lambda_q \cdot \sigma_{\max}^2(\mathbf{W}^Q) + \lambda_k \cdot \sigma_{\max}^2(\mathbf{W}^K) + \lambda_v \cdot \sigma_{\max}^2(\mathbf{W}^V), \quad (13)$$

where $\lambda_q, \lambda_k$ and $\lambda_v$ are trade-off parameters. The original Transformer Vaswani et al. (2017) adopts multi-head self-attention to jointly attend to information from different subspaces at different positions. In line with that spirit, MSVP can also be added in a multi-head manner. Incorporating the summation over all the heads and layers, the overall training objective with multi-head MSVP is:

$$\mathcal{J} = \mathcal{L}_{cls} + \mathcal{L}_{msvp} = \mathcal{L}_{cls} + \lambda_q \cdot \sum_{l=1}^{L}\sum_{h=1}^{H} \sigma_{\max}^2(\mathbf{W}_l^{Q,h}) + \lambda_k \cdot \sum_{l=1}^{L}\sum_{h=1}^{H} \sigma_{\max}^2(\mathbf{W}_l^{K,h}) + \lambda_v \cdot \sum_{l=1}^{L}\sum_{h=1}^{H} \sigma_{\max}^2(\mathbf{W}_l^{V,h}), \quad (14)$$

where $\mathbf{W}_l^{*,h}$ denotes the $h^{th}$ head in the $l^{th}$ attention layer, and $*$ could be $Q, K$ or $V$. By restricting the maximum size of the perturbation's largest singular value, we can control the range of the output changes when attacked, leading to a model with better robustness. The computation of maximum singular values can be seamlessly integrated into the forward process.

## B  PROOF OF THEOREM 4.3

In this section, we derive the local Lipschitz constant upper bound for the self-attention mechanism (**Attn**) around input $\mathbf{X}_0$.

**Theorem 3.2** (Local Lipschitz Constant of self-attention layer). *The self-attention layer is local Lipschitz continuous in $B_2(\mathbf{X}_0, \delta_0)$*

$$\mathrm{Lip}_2(\mathrm{Attn}, \mathbf{X}_0) \leq N(N+1)(B+\delta_0)^2 \left[ \left\| \mathbf{W}^V \right\|_2 \left\| \mathbf{W}^Q \right\|_2 \left\| \mathbf{W}^{K,\top} \right\|_2 + \left\| \mathbf{W}^V \right\|_2 \right]. \quad (15)$$

*Proof.* To begin, we introduce some fundamental notations and re-formulate the self-attention mechanism as follows:

$$\begin{aligned} \mathrm{Attn}\left(\mathbf{X}, \mathbf{W}^Q, \mathbf{W}^K, \mathbf{W}^V\right) &= \mathrm{softmax}\left( \frac{\mathbf{X}\mathbf{W}^Q \left(\mathbf{X}\mathbf{W}^K\right)^\top}{\sqrt{D}} \right) \mathbf{X}\mathbf{W}^V, \\ &= \mathbf{P}\mathbf{X}\mathbf{W}^V, \end{aligned} \quad (16)$$

where we use $\mathbf{P}$ to denote the softmax matrix for brevity. We can further express the attention mechanism as a collection of row vector functions $f_i(\mathbf{X})$:

$$f(\mathbf{X}) = \mathrm{Attn}(\mathbf{X}) = \mathbf{P}\mathbf{X}\mathbf{W}^V = \begin{bmatrix} f_1(\mathbf{X}) \\ \vdots \\ f_N(\mathbf{X}) \end{bmatrix} \in \mathbb{R}^{N \times D}, \text{ where } f_i(\mathbf{X}) \in \mathbb{R}^{1 \times D}, \quad (17)$$

We denote the input data matrix $\mathbf{X}$ as a collection of row vectors and the $(i, j)$ element of $\mathbf{P}$ as $P_{ij}$:

$$\mathbf{X} = \begin{bmatrix} \mathbf{x}_1^\top \\ \vdots \\ \mathbf{x}_N^\top \end{bmatrix} \in \mathbb{R}^{N \times d}, \text{ where } \mathbf{x}_i^\top \in \mathbb{R}^{1 \times d},$$

Then we have the representation for the row vector function $f_i(\mathbf{X})$:

$$f_i(\mathbf{X}) = \sum_{j=1}^{N} P_{ij}\mathbf{x}_j^\top \mathbf{W}^V, \quad (18)$$

Denote the transpose of the $i^{th}$ row of the softmax matrix $\mathbf{P}$ as $\mathbf{P}_{i:}^{\top} = \mathrm{softmax}(\mathbf{XAx}_i) \in \mathbb{R}^{N \times 1}$, where $\mathbf{A}$ stands for $\frac{\mathbf{W}^Q \mathbf{W}^{K,\top}}{\sqrt{D}}$ , we can further write it as a matrix multiplication form:

$$f_i(\mathbf{X}) = \mathbf{P}_{i:}\mathbf{XW}^V \in \mathbb{R}^{1 \times D}, \tag{19}$$

To calculate the Jacobian of the attention mechanism, we need to introduce some preliminaries. Since $f$ is a map from $\mathbb{R}^{N \times D}$ to $\mathbb{R}^{N \times d}$, its Jacobian is

$$\mathbf{J}_f(\mathbf{X}) = \begin{bmatrix} \mathbf{J}_{11}(\mathbf{X}) & \dots & \mathbf{J}_{1N}(\mathbf{X}) \\ \vdots & \ddots & \vdots \\ \mathbf{J}_{N1}(\mathbf{X}) & \dots & \mathbf{J}_{NN}(\mathbf{X}) \end{bmatrix} \in \mathbb{R}^{ND \times Nd}, \tag{20}$$

where $\mathbf{J}_{ij}(\mathbf{X}) = \frac{\partial f_i(\mathbf{X})}{\partial \mathbf{x}_j} \in \mathbb{R}^{D \times d}$.

Recall that

$$f_i(\mathbf{X}) = \sum_{j=1}^{N} P_{ij}\mathbf{x}_j^{\top}\mathbf{W}^V = \mathbf{P}_{i:}\mathbf{XW}^V \in \mathbb{R}^{1 \times D}, \tag{21}$$

where $P_{ij}$ is a function that depends on $\mathbf{X}$: $\mathbf{P}_{i:}^{\top} = \mathrm{softmax}(\mathbf{XAx}_i) \in \mathbb{R}^{N \times 1}$. Denote $\mathbf{XAx}_i = \mathbf{q}$ for brevity, by applying the chain rule and product rule, we obtain a commonly used result that can be applied as follows:

$$\frac{\partial \mathbf{P}_{i:}}{\partial \mathbf{q}} = \frac{\partial \, \mathrm{softmax}(\mathbf{q})}{\partial \mathbf{q}} = \mathrm{diag}(\mathbf{P}_{i:}) - \mathbf{P}_{i:}\mathbf{P}_{i:}^{\top} := \mathbf{P}^{(i)}, \tag{22}$$

By utilizing this result, we can continue to calculate the Jacobian,

$$\frac{\partial f_i(\mathbf{X})}{\partial \mathbf{x}_j} = \sum_{k=1}^{N} \mathbf{W}^V \left( \frac{\partial P_{ik}}{\partial \mathbf{x}_j}\mathbf{x}_k + P_{ik}\frac{\partial \mathbf{x}_k}{\partial \mathbf{x}_j} \right),$$

$$= \mathbf{W}^V \sum_{k=1,k \neq j}^{N} \frac{\partial P_{ik}}{\partial \mathbf{x}_j}\mathbf{x}_k + \mathbf{W}^V \frac{\partial P_{ij}}{\partial \mathbf{x}_j}\mathbf{x}_j + \mathbf{W}^V P_{ij}I, \tag{23}$$

$$= \mathbf{W}^V \sum_{k=1}^{N} \frac{\partial P_{ik}}{\partial \mathbf{x}_j}\mathbf{x}_k + \mathbf{W}^V P_{ij}I,$$

We can further write it into a matrix form for the following derivation:

$$\frac{\partial f_i(\mathbf{X})}{\partial \mathbf{x}_j} = \mathbf{W}^V \sum_{k=1}^{N} \frac{\partial P_{ik}}{\partial \mathbf{x}_j}\mathbf{x}_k + \mathbf{W}^V P_{ij}I = \mathbf{W}^V \left[\mathbf{x}_1 \dots \mathbf{x}_N\right] \begin{bmatrix} \frac{\partial P_{i1}}{\partial \mathbf{x}_j} \\ \vdots \\ \frac{\partial P_{iN}}{\partial \mathbf{x}_j} \end{bmatrix} + \mathbf{W}^V P_{ij}I,$$

$$= \mathbf{W}^V \mathbf{X}^{\top} \begin{bmatrix} \frac{\partial P_{i1}}{\partial \mathbf{x}_j} \\ \vdots \\ \frac{\partial P_{iN}}{\partial \mathbf{x}_j} \end{bmatrix} + \mathbf{W}^V P_{ij}I = \mathbf{W}^V \left( \underbrace{\mathbf{X}^{\top}\frac{\partial \mathbf{P}_{i:}^{\top}}{\partial \mathbf{x}_j}}_{(I)} + \underbrace{P_{ij}I}_{(II)} \right), \tag{24}$$

By utilizing the chain rule again and with the result from equation (22), we can further decompose the first term (I) as

$$\mathbf{X}^{\top}\frac{\partial \mathbf{P}_{i:}^{\top}}{\partial \mathbf{x}_j} = \mathbf{X}^{\top}\frac{\partial \mathbf{P}_{i:}^{\top}}{\partial \mathbf{q}}\frac{\partial \mathbf{q}}{\partial \mathbf{x}_j} = \mathbf{X}^{\top}\mathbf{P}^{(i)}\frac{\partial \mathbf{q}}{\partial \mathbf{x}_j}, \tag{25}$$

Recall that $\mathbf{q} = \mathbf{XAx}_i$, we can analyze $\frac{\partial \mathbf{q}}{\partial \mathbf{x}_j} = \frac{\partial \mathbf{XAx}_i}{\partial \mathbf{x}_j}$ in two cases:

- Case 1: if $i \neq j$,

$$\frac{\partial \mathbf{XAx}_i}{\partial \mathbf{x}_j} = \begin{bmatrix} \left(\frac{\partial \mathbf{x}_1^{\top}\mathbf{Ax}_i}{\partial \mathbf{x}_j}\right)^{\top} \\ \vdots \\ \left(\frac{\partial \mathbf{x}_N^{\top}\mathbf{Ax}_i}{\partial \mathbf{x}_j}\right)^{\top} \end{bmatrix} = \begin{bmatrix} \mathbf{0}^{\top} \\ .. \\ (\mathbf{Ax}_i)^{\top} \\ .. \\ \mathbf{0}^{\top} \end{bmatrix} = \begin{bmatrix} \mathbf{0}^{\top} \\ .. \\ \mathbf{x}_i^{\top}\mathbf{A}^{\top} \\ .. \\ \mathbf{0}^{\top} \end{bmatrix} = \begin{bmatrix} \mathbf{0}^{\top} \\ .. \\ \mathbf{x}_i^{\top} \\ .. \\ \mathbf{0}^{\top} \end{bmatrix} \mathbf{A}^{\top} = \mathbf{E}_{ji}\mathbf{XA}^{\top},$$

$$\tag{26}$$

where $\mathbf{E}_{ji}$ represents the all-zero matrix except for the entry at $(j,i)$, which is equal to 1.

- Case 2: if $i = j$,

$$\frac{\partial \mathbf{X}\mathbf{A}\mathbf{x}_i}{\partial \mathbf{x}_i} = \begin{bmatrix} \left(\frac{\partial \mathbf{x}_1^\top \mathbf{A}\mathbf{x}_i}{\partial \mathbf{x}_i}\right)^\top \\ \vdots \\ \left(\frac{\partial \mathbf{x}_N^\top \mathbf{A}\mathbf{x}_i}{\partial \mathbf{x}_i}\right)^\top \end{bmatrix}, \tag{27}$$

Note that for $k \neq i$, we have

$$\frac{\partial \mathbf{x}_k^\top \mathbf{A}\mathbf{x}_i}{\partial \mathbf{x}_i} = \left(\mathbf{x}_k^\top \mathbf{A}\right)^\top = \mathbf{A}^\top \mathbf{x}_k, \tag{28}$$

when $k = i$, we have

$$\frac{\partial \mathbf{x}_i^\top \mathbf{A}\mathbf{x}_i}{\partial \mathbf{x}_i} = \mathbf{A}\mathbf{x}_i + \left(\mathbf{x}_i^\top \mathbf{A}\right)^\top = \mathbf{A}\mathbf{x}_i + \mathbf{A}^\top \mathbf{x}_i, \tag{29}$$

Therefore,

$$\begin{aligned} \frac{\partial \mathbf{X}\mathbf{A}\mathbf{x}_i}{\partial \mathbf{x}_i} &= \begin{bmatrix} \left(\frac{\partial \mathbf{x}_1^\top \mathbf{A}\mathbf{x}_i}{\partial \mathbf{x}_i}\right)^\top \\ \vdots \\ \left(\frac{\partial \mathbf{x}_N^\top \mathbf{A}\mathbf{x}_i}{\partial \mathbf{x}_i}\right)^\top \end{bmatrix} = \begin{bmatrix} (\mathbf{A}^\top \mathbf{x}_1)^\top \\ .. \\ (\mathbf{A}\mathbf{x}_i + \mathbf{A}^\top \mathbf{x}_i)^\top \\ .. \\ (\mathbf{A}^\top \mathbf{x}_N)^\top \end{bmatrix} = \begin{bmatrix} \mathbf{x}_1^\top \mathbf{A}^\top \\ .. \\ \mathbf{x}_i^\top \mathbf{A} + \mathbf{x}_i^\top \mathbf{A}^\top \\ .. \\ \mathbf{x}_N^\top \mathbf{A}^\top \end{bmatrix}, \\ &= \begin{bmatrix} \mathbf{0}^\top \\ .. \\ \mathbf{x}_i^\top \mathbf{A}^\top \\ .. \\ \mathbf{0}^\top \end{bmatrix} + \mathbf{X}\mathbf{A} = \mathbf{E}_{ii}\mathbf{X}\mathbf{A}^\top + \mathbf{X}\mathbf{A}, \end{aligned} \tag{30}$$

By combining the two cases above, we can express $\frac{\partial \mathbf{q}}{\partial \mathbf{x}_j}$ using a unified formula:

$$\frac{\partial \mathbf{q}}{\partial \mathbf{x}_j} = \mathbf{E}_{ji}\mathbf{X}\mathbf{A}^\top + \mathbf{X}\mathbf{A}\delta_{ij}, \tag{31}$$

where $\delta_{ij}$ is the Kronecker delta function, which takes the value 1 when $i = j$ and 0 otherwise. Combining equation 25, 31, we have

$$\mathbf{X}^\top \frac{\partial \mathbf{P}_{i:}^\top}{\partial \mathbf{x}_j} = \mathbf{X}^\top \frac{\partial \mathbf{P}_{i:}^\top}{\partial \mathbf{q}}\frac{\partial \mathbf{q}}{\partial \mathbf{x}_j} = \mathbf{X}^\top \mathbf{P}^{(i)}\frac{\partial \mathbf{q}}{\partial \mathbf{x}_j} = \mathbf{X}^\top \mathbf{P}^{(i)}(\mathbf{E}_{ji}\mathbf{X}\mathbf{A}^\top + \mathbf{X}\mathbf{A}\delta_{ij}), \tag{32}$$

Substituting this result back into equation 24, we can get the $(i,j)$ block of the Jacobian:

$$\begin{aligned} \mathbf{J}_{ij}(\mathbf{X}) = \frac{\partial f_i(\mathbf{X})}{\partial \mathbf{x}_j} &= \mathbf{W}^V \left(\mathbf{X}^\top \frac{\partial \mathbf{P}_{i:}^\top}{\partial \mathbf{x}_j} + P_{ij}I\right), \\ &= \mathbf{W}^V \left(\mathbf{X}^\top \mathbf{P}^{(i)}(\mathbf{E}_{ji}\mathbf{X}\mathbf{A}^\top + \mathbf{X}\mathbf{A}\delta_{ij}) + P_{ij}I\right), \quad (\forall 1 \leq i, j \leq N) \end{aligned} \tag{33}$$

Specifically, we can write the diagnol $(i,i)$ block of the Jacobian as

$$\mathbf{J}_{ii}(\mathbf{X}) = \mathbf{W}^V \left(\mathbf{X}^\top \mathbf{P}^{(i)}(\mathbf{E}_{ii}\mathbf{X}\mathbf{A}^\top + \mathbf{X}\mathbf{A}) + P_{ii}I\right), \tag{34}$$

while the non-diagnoal $(i,j), j \neq i$ block can be written as

$$\mathbf{J}_{ij}(\mathbf{X}) = \mathbf{W}^V \left(\mathbf{X}^\top \mathbf{P}^{(i)}\mathbf{E}_{ij}\mathbf{X}\mathbf{A}^\top + P_{ij}I\right), \tag{35}$$

Let us denote the $i^{th}$ block row of the Jacobian $\mathbf{J}$ as $[\mathbf{J}_{i1}, \cdots, \mathbf{J}_{iN}]$. We state the following lemma, which establishes a connection between the spectral norm of a block matrix and its block rows:

**Lemma B.1** (Relationship between the spectral norm of a block row and the block matrix, Kim et al. (2021)). *Let $\mathbf{A}$ be a block matrix with block columns $\mathbf{A}_1, \ldots, \mathbf{A}_N$. Then $\|A\|_2 \leq \sqrt{\sum_i \|\mathbf{A}_i\|_2^2}$.*

Utilizing this lemma, and consider the input around $\mathbf{X}_0$: $B_2(\mathbf{X}_0, \delta_0) := \{\mathbf{X} : \|\mathbf{X} - \mathbf{X}_0\|_F \leq \delta_0\}$, we can focus on derive the spectral norm of the $i^{th}$ block row $[\mathbf{J}_{i1}, \cdots, \mathbf{J}_{iN}]$ in the neighbourhood. Considering that all inputs received by ViT are bounded (for instance, all entries of image data fall within the range of 0 to 255), and the inputs entering the Attention layer have been normalized by LayerNorm to a specific range $[0, 1]$, we can further replace the norm $\|\mathbf{X}\|$ with a constant $B = \max_{\mathbf{X} \in \mathcal{X}} \|\mathbf{X}\|_2$, where $\mathcal{X}$ represents a bounded open set in the Euclidean space $\mathbb{R}^{N \times D}$.

$$
\begin{aligned}
&\|[\mathbf{J}_{i1}(\mathbf{X}), \ldots, \mathbf{J}_{iN}(\mathbf{X})]\|_2 \Big|_{\mathbf{X} \in B_2(\mathbf{X}_0, \delta_0)} \\
&\leq \|\mathbf{J}_{ii}(\mathbf{X})\|_2 + \sum_{j \neq i} \|\mathbf{J}_{ij}(\mathbf{X})\|_2 \Big|_{\mathbf{X} \in B_2(\mathbf{X}_0, \delta_0)}, \\
&= \|\mathbf{W}^V (\mathbf{X}^\top \mathbf{P}^{(i)} (\mathbf{E}_{ii} \mathbf{X} \mathbf{A}^\top + \mathbf{X} \mathbf{A}) + P_{ii} I)\|_2 + \sum_{j \neq i} \|\mathbf{W}^V (\mathbf{X}^\top \mathbf{P}^{(i)} \mathbf{E}_{ij} \mathbf{X} \mathbf{A}^\top + P_{ij} I)\|_2 \\
&\leq \|\mathbf{W}^V\|_2 \|\mathbf{P}^{(i)}\|_2 (\|\mathbf{E}_{ii}\|_2 \|\mathbf{W}^Q\|_2 \|\mathbf{W}^{K,\top}\|_2 + \|\mathbf{W}^Q\|_2 \|\mathbf{W}^{K,\top}\|) (B + \delta_0)^2 + \|\mathbf{W}^V\|_2, \\
&\sum_{j \neq i} [\|\mathbf{W}^V\|_2 \|\mathbf{P}^{(i)}\|_2 \|\mathbf{E}_{ij}\|_2 \|\mathbf{W}^Q\|_2 \|\mathbf{W}^{K,\top}\|_2 + \|\mathbf{W}^V\|_2] (B + \delta_0)^2,
\end{aligned}
\tag{36}
$$

The first equality is in accordance with equation 34,35, while the second inequality arises from the Cauchy-Schwarz inequality and the boundedness of the input space $\mathcal{X}$ and the perturbation $\delta_0$. Furthermore, by utilizing $\|\mathbf{P}^{(i)}\|_2 \leq 1$ and $\|\mathbf{E}_{ij}\|_2 \leq 1$, we can omit them in the inequality, leave with pure weight matrices,

$$
\begin{aligned}
&\|[\mathbf{J}_{i1}(\mathbf{X}), \ldots, \mathbf{J}_{iN}(\mathbf{X})]\|_2 \Big|_{\mathbf{X} \in B_2(\mathbf{X}_0, \delta_0)} \\
&\leq 2(B + \delta_0)^2 \|\mathbf{W}^V\|_2 (\|\mathbf{W}^Q\|_2 \|\mathbf{W}^{K,\top}\|_2 + 1) + \\
&\quad (N - 1)(B + \delta_0)^2 [\|\mathbf{W}^V\|_2 \|\mathbf{W}^Q\|_2 \|\mathbf{W}^{K,\top}\|_2 + \|\mathbf{W}^V\|_2], \\
&= (N + 1)(B + \delta_0)^2 [\|\mathbf{W}^V\|_2 \|\mathbf{W}^Q\|_2 \|\mathbf{W}^{K,\top}\|_2 + \|\mathbf{W}^V\|_2],
\end{aligned}
\tag{37}
$$

By directly summing across the row index or utilizing the lemma B.1 can yield our main theorem.

$$
\|\mathbf{J}\|_2 \leq N(N + 1)(B + \delta_0)^2 [\|\mathbf{W}^V\|_2 \|\mathbf{W}^Q\|_2 \|\mathbf{W}^{K,\top}\|_2 + \|\mathbf{W}^V\|_2].
\tag{38}
$$

This ends the proof.

$\square$

## C  PROOF OF THEOREM 5.1

**Theorem 4.1** (Convergence guarantee of the power iteration method (Mises & Pollaczek-Geiringer, 1929)). *Assuming the dominant singular value $\sigma_{max}(\mathbf{A})$ (which is also the eigenvalue of matrix $\mathbf{A}^\top \mathbf{A}$) is strictly greater than the subsequent singular values and that $\mathbf{u}_0$ is initially selected at random, then there is a probability of 1 that $\mathbf{u}_0$ will have a non-zero component in the direction of the eigenvector linked with the dominant singular value. Consequently, the convergence will be geometric with a ratio of $\left| \frac{\sigma_2(\mathbf{A})}{\sigma_{max}(\mathbf{A})} \right|$.*

*Proof.* Denote $\sigma_1, \sigma_2, \cdots, \sigma_m$ as the $m$ eigenvalues of matrix $\mathbf{A}^\top \mathbf{A}$, and $\mathbf{v}_1, \mathbf{v}_2, \cdots, \mathbf{v}_m$ as the corresponding eigenvectors. Suppose $\sigma_1$ is the dominant eigenvalue, denote as $\sigma_{max} = \sigma_1$, with

$|\sigma_1| > |\sigma_j|$ for $\forall j > 1$. The initial vector $\boldsymbol{u}_0$ can be written as the linear combination of the eigenvectors: $\boldsymbol{u}_0 = c_1 \mathbf{v}_1 + c_2 \mathbf{v}_2 + \cdots + c_m \mathbf{v}_m$. If $\boldsymbol{u}_0$ is chosen randomly with uniform probability, then the eigenvector corresponding to the largest singular value has a nonzero coefficient, namely $c_1 \neq 0$. After multiplying the initial vector $\boldsymbol{u}_0$ with the matrix $\mathbf{A}$ $k$ times, we have:

$$
\begin{aligned}
\mathbf{A}^k \boldsymbol{u}_0 &= c_1 \mathbf{A}^k \mathbf{v}_1 + c_2 \mathbf{A}^k \mathbf{v}_2 + \cdots + c_m \mathbf{A}^k \mathbf{v}_m, \\
&= c_1 \sigma_1^k \mathbf{v}_1 + c_2 \sigma_2^k \mathbf{v}_2 + \cdots + c_m \sigma_m^k \mathbf{v}_m, \\
&= c_1 \sigma_1^k \left( \mathbf{v}_1 + \frac{c_2}{c_1} \left( \frac{\sigma_2}{\sigma_1} \right)^k \mathbf{v}_2 + \cdots + \frac{c_m}{c_1} \left( \frac{\sigma_m}{\sigma_1} \right)^k \mathbf{v}_m \right), \\
&\rightarrow c_1 \sigma_1^k \mathbf{v}_1 \quad (k \rightarrow \infty).
\end{aligned}
\tag{39}
$$

The second equality holds for the eigenvectors with $\mathbf{A}^k \mathbf{v}_i = \sigma_i^k \mathbf{v}_i$, $\forall i = 1, \cdots, m$, while the last equality is valid when $\left| \frac{\sigma_i}{\sigma_1} \right| < 1$ for all $i > 1$.

On the other hand, the vector for iterative step $k$ can be written as $\boldsymbol{u}_k = \frac{A^k \boldsymbol{u}_0}{\|A^k \boldsymbol{u}_0\|}$. Combining these two equations above, we obtain $\boldsymbol{u}_k \rightarrow C \mathbf{v}_1$ as $k \rightarrow \infty$, where $C$ is a constant. Therefore, we can use the power iteration method to approximate the largest singular value. The convergence is geometric, with a ratio of $\left| \frac{\sigma_2}{\sigma_1} \right|$. $\qquad \square$

## D   COMPARISON WITH EXISTING BOUNDS

Existing literature on introducing Lipschitz continuity into Transformer models comprises three notable articles(Kim et al., 2021; Dasoulas et al., 2021; Qi et al., 2023), we abbreviate their proposed modified Transformer as L2Former(Kim et al., 2021), LNFormer(Dasoulas et al., 2021), and Lips-Former(Qi et al., 2023). We summarize the three bounds they propose for their modified attention mechanisms below. We have also included the vanilla self-attention mechanism, along with these three modified attention mechanisms and our optimization objectives, in Table 4, where we examine whether these studies address robustness and whether they simplify the analysis of mechanisms.

TABLE 4: Comprehensive Comparison of the Baseline Models.

| Model | Proposed Mechanism | Lipschitz Continuity | Robustness | Simplified |
|---|---|---|---|---|
| Transformer (Dosovitskiy et al., 2020) | $\text{softmax}\left( \frac{\mathbf{X}\mathbf{W}^Q(\mathbf{X}\mathbf{W}^K)^\top}{\sqrt{D}} \right) \mathbf{X}\mathbf{W}^V$ | × | × | − |
| L2Former (Kim et al., 2021) | $\exp\left( -\frac{\left\| \mathbf{x}_i^\top W^Q - \mathbf{x}_j^\top W^K \right\|_2^2}{\sqrt{D/H}} \right)$ | ✓ | × | ✓ |
| LNFormer (Dasoulas et al., 2021) | $\frac{Q^\top K}{\max\{uv, uw, vw\}}$ | ✓ | × | ✓ |
| LipsFormer (Qi et al., 2023) | $\mathbf{q}_i = \frac{(\mathbf{x}_i^\top \mathbf{W}^Q)^\top}{\sqrt{\|\mathbf{x}_i^\top \mathbf{W}^Q\|^2 + \epsilon}}$ | ✓ | × | × |
| SpecFormer (Ours) | $\mathcal{L}_{cls} + \lambda \cdot \sigma_{\max}^2(\mathbf{W})$ | ✓ | ✓ | × |

In (Kim et al., 2021), it was demonstrated that the self-attention mechanism, due to the potential unboundedness of its inputs, is not globally Lipschitz continuous. Instead, they introduced an L2 self-attention mechanism based on the L2 distance $\left\| \mathbf{x}_i^\top W^Q - \mathbf{x}_j^\top W^K \right\|_2^2$. However, their proposed L2 attention mechanism also lacks global Lipschitz continuity in cases where $\mathbf{W}^K \neq \mathbf{W}^Q$, possessing global Lipschitz continuity only when $\mathbf{W}^K = \mathbf{W}^Q$. This constraint imposes significant limitations on various expressive capabilities of the model itself, as demonstrated by the experimental results presented in the original paper's Appendix L. Confining the model to $\mathbf{W}^K = \mathbf{W}^Q$ results in a substantially higher loss during convergence compared to the unconstrained model, rendering the model suboptimal.

In (Dasoulas et al., 2021), the authors proposed a normalization approach, denoted as $g(\mathbf{X}) = \frac{\tilde{g}(\mathbf{X})}{c(\mathbf{X})}$, where $g(X)$ represents the score function in the softmax operation, to address the issue of potentially unbounded inputs, as mentioned earlier. However, their analysis is based on a simplified version of the attention mechanism, which significantly differs from the practical application of the self-attention mechanism. Additionally, through their choice of the function $c(\mathbf{X})$, it appears that the authors have implicitly assumed that the input is bounded. They defined $c(\mathbf{X})$ as follows: $c(\mathbf{X}) = \max\{\|\mathbf{Q}\|_F \|\mathbf{K}^\top\|_{(\infty,2)}, \|\mathbf{Q}\|_F \|\mathbf{V}^\top\|_{(\infty,2)}, \|\mathbf{K}^\top\|_{(\infty,2)} \|\mathbf{V}^\top\|_{(\infty,2)}\}$. Here, the input matrix $\mathbf{X}$ is represented as $\mathbf{X} = (\mathbf{Q}\|\mathbf{K}\|\mathbf{V})$. It's worth noting that this choice of $c(\mathbf{X})$ seems to assume boundedness in the input data, which may not fully address the issue of unbounded inputs in practical self-attention mechanisms.

In (Qi et al., 2023), the authors introduced a series of modules to replace components in the vanilla Vision Transformer that they deemed to introduce instability during training. These modifications include using CenterNorm to replace LayerNorm, employing scaled cosine similarity attention (SCSA) as an alternative to vanilla self-attention, and utilizing weighted residual shortcuts controlled by additional learnable parameters, along with spectral initialization for convolutions and feed-forward connections, all aimed at ensuring that the Lipschitz constant for each component remains below 1 for training stability.

However, it is noteworthy that these improvements, as presented, appear unnecessary and excessively restrictive in terms of the model's expressiveness. For instance, the authors created CenterNorm and SCSA by merely shifting the operation of dividing by the standard deviation from LayerNorm to the attention layer, resulting in no substantial improvement. Furthermore, the $\ell_2$ row normalization applied to the $\mathbf{Q}, \mathbf{K}, \mathbf{V}$ matrices strongly impacts the overall expressiveness of the model, imposing overly stringent constraints.

Additionally, the introduction of the extra parameter alpha in the weighted residual shortcut to control the influence of the residual path on the entire pathway, while aiming for contractive Lipschitz properties, restricts $\alpha$ to be learned within a predefined, highly limited range or even kept fixed. These operations are cumbersome and introduce additional computational overhead.

Lastly, the proposed spectral initialization, though effective in ensuring a Lipschitz constant of 1 for convolutions and feed-forward parts, is extremely time-consuming during the initialization phase. Despite these extensive operations, the SCSA module in the authors' paper still requires **bounded norm** values for $\mathbf{W}^Q, \mathbf{W}^K, \mathbf{W}^V$ to satisfy Lipschitz properties. Our observation posits that, in practical scenarios, due to the Lipschitz continuity of LayerNorm, each entry of the input image (e.g., pixel values ranging between 0 and 255) is inherently bounded. Moreover, the LayerNorm applied to the input before entering the attention layer constrains the input values to the $[0, 1]$ range. Therefore, when contemplating the Lipschitz continuity of the attention layer in practical terms, it suffices to consider bounded conditions. To delve further, in the context of adversarial robustness, we only need to consider pointwise Lipschitz continuity and, more specifically, the local Lipschitz continuity around a fixed point. This Lipschitz continuity can be reliably guaranteed.

# E  EXPERIMENTS

## E.1  HYPERPARAMETER ANALYSIS

In this section, we present the results of a hyperparameter ablation study. From Table 5, it is evident that the performance of our proposed SpecFormer remains robust even with varying hyperparameters. We recommend a tuning range of $[1e-6, 1e-3]$ for optimal results. When penalties that are too large, such as $1e-2$, can result in overly constrained representations, whereas penalties that are too small, such as $1e-7$, may have little to no effect.

## E.2  IMPLEMENTATION DETAILS

The experiments were conducted on a system equipped with an Intel(R) Xeon(R) CPU E5-2687W v4 @ 3.00GHz and NVIDIA TITAN RTX. The abbreviation ViT-S stands for the Vision Transformer (Dosovitskiy et al., 2020) Small backbone, which comprises of 8 attention blocks with 8 heads and an embedding dimension of 768. The DeiT-Ti backbone, on the other hand, refers to the DeiT (Touvron et al., 2021a) Tiny model, which comprises of 12 attention blocks with 3 heads and

TABLE 5: Performance (%) of SpecFormer on CIFAR10 under different hyperparameter choices for both standard training and adversarial training. The best results are highlighted in **bold**.

| $(\lambda_q, \lambda_k, \lambda_v)$ | CIFAR-10-Standard Training | | | CIFAR-10-Adversarial Training | | |
|---|---|---|---|---|---|---|
| | Standard | PGD-2 | FGSM | Standard | CW-20 | PGD-20 |
| (1e-4,0,0) | 87.36 | 35.32 | **53.91** | 72.54 | 31.47 | 31.59 |
| (0,1e-4,0) | 88.58 | 29.02 | 49.81 | **72.56** | 31.37 | 31.19 |
| (0,0,1e-4) | 87.66 | **36.19** | 52.40 | 72.51 | 31.41 | 31.65 |
| (5e-4, 7e-5 ,2e-4) | 88.32 | 31.42 | 48.82 | 72.53 | 31.59 | 31.92 |
| (1e-4,1e-4,1e-4) | 88.14 | 36.07 | 53.71 | 72.46 | **31.94** | 32.23 |
| (2e-4,5e-5,7e-5) | 88.61 | 30.01 | 48.31 | 72.42 | 31.56 | 31.82 |
| (1e-3,9e-5,3e-4) | 88.15 | 31.75 | 50.46 | 72.43 | 31.91 | 32.23 |
| (1e-3,1e-3,1e-3) | 88.07 | 32.59 | 47.85 | 71.70 | 31.31 | 31.79 |
| (5e-4,3e-4,3e-4) | **88.81** | 31.28 | 48.32 | 72.19 | 31.69 | 31.89 |
| (5e-5,3e-5,3e-5) | 88.52 | 29.53 | 50.58 | 72.55 | 31.68 | 31.72 |
| (2e-3,3e-4,4e-4) | 88.29 | 33.25 | 48.25 | 71.75 | 31.75 | **32.29** |
| (4e-3,8e-4,1e-3) | 88.29 | 33.94 | 46.72 | 71.37 | 31.84 | 32.19 |

TABLE 6: Performance (%) of SpecFormer with different ViT variants on ImageNet datasets under standard training (using ImageNet-22k pre-trained weights). The best results are in **bold**.

| Model | Method | Standard Training | | | Adversarial Training | | | |
|---|---|---|---|---|---|---|---|---|
| | | Standard | FGSM | PGD-2 | Standard | CW-20 | PGD-20 | PGD-100 |
| | LipsFormer (Qi et al., 2023) | 65.76 | 20.87 | 3.77 | 45.04 | 18.91 | 21.13 | 20.83 |
| | L2Former (Kim et al., 2021) | 77.40 | 37.12 | 6.89 | 51.24 | 24.85 | 27.04 | 26.95 |
| ViT-B | LNFormer (Dasoulas et al., 2021) | 50.84 | 25.78 | 0.54 | 30.93 | 11.53 | 14.08 | 14.04 |
| | Transformer (Dosovitskiy et al., 2020) | 79.11 | 41.45 | 10.79 | 60.81 | 30.92 | 32.58 | 32.35 |
| | SpecFormer (Ours) | **80.04** | **43.51** | **11.59** | **62.30** | **31.87** | **32.82** | **32.56** |

an embedding dimension of 192. Finally, ConViT-Ti refers to the ConViT (d'Ascoli et al., 2021) Tiny model, comprising of 10 layers with 4 heads and an embedding size of 48.

### E.3   EVALUATION ON IMAGENET

We assess the performance of our SpecFormer model on ImageNet-1k, a widely used large-scale dataset. We conduct a comparative analysis of SpecFormer against four other baseline models under both standard training and adversarial training paradigms. For standard training, we evaluate the model's adversarial robustness using FGSM (Goodfellow et al., 2015) and PGD-2 (Madry et al., 2018) attacks, employing an attack radius of $2/255$. In the case of adversarially-trained models, we rigorously evaluate their robustness against CW-20 (Carlini & Wagner, 2017) as well as PGD-20 and PGD-100 (Madry et al., 2018) attacks, utilizing an attack radius of $8/255$. Following the training protocols outlined in (Mo et al., 2022), we conducted adversarial training of the models for a total of 10 epochs. Our learning rate schedule aligns with their implementation, involving a reduction of the learning rate by a factor of 10 at the $6^{th}$ and $8^{th}$ epochs. The experimental results on ImageNet are presented in Table 6. It is evident from the table that SpecFormer outperforms the other models under both standard and adversarial training paradigms, achieving the highest standard and robust accuracy. This highlights the effectiveness and superiority of our approach on a large-scale dataset.

