# OpenReview forum: "SpecFormer: Guarding Vision Transformer Robustness via Maximum Singular Value Penalization"
_ICLR.cc/2024/Conference — ICLR 2024 Conference Withdrawn Submission_

### Official Review · Reviewer_B7sA · 2023-10-26

**Soundness:** 4 excellent
**Presentation:** 3 good
**Contribution:** 4 excellent
**Rating:** 8
**Confidence:** 5

**Summary:**

This paper proposes SpecFormer, a method to improve the adversarial robustness of Vision Transformer (ViT) models. The authors first provide a theoretical analysis of the local Lipschitz continuity of the self-attention mechanism, and establish its connection with the robustness of ViT models. Based on this, they propose a Maximum Singular Value Penalization (MSVP) algorithm that can effectively control the local Lipschitz constant of attention layers to improve model robustness.

**Strengths:**

The novel formulation of studying ViT robustness from the perspective of local Lipschitz continuity provides an insightful theoretical framework. Establishing this connection between self-attention mechanisms and model robustness is a creative combination of existing theories that opens up new possibilities.
The proposed MSVP algorithm is simple, novel, yet ingenious. By seamlessly integrating spectral control into ViT self-attention, it enhances robustness with minimal overhead. This creative application in a new domain removes limitations of prior empirical defenses.
Experiments on various datasets and models solidly verify the efficacy and consistency of the methods. The superior performance shows this is a significant advancement with real-world impact.
The writing is lucid, and the derivations are meticulous. The theoretical analysis firmly grounds the technical innovation on solid mathematical principles. The paper quality is publication-worthy.

**Weaknesses:**

Some steps in the proof are a little bit concise that are not easy to understand. For instance, in the proof of Theorem 4.3, can you explain in detail the steps involved in utilizing Cauchy’s inequality and boundedness of input space to arrive at inequality (36)? Adding these intermediate deductions would help enhance the clarity of the theoretical derivations.
More visual results, such as attention maps, can be provided to further visually demonstrate the impact of the proposed method on safeguarding the attention mechanism under malicious attacks.

**Questions:**

Can the method be easily applied to other real-life datasets such as surveillance videos, facial recognition, or remote sensing images to enhance vision safety, in addition to experimental datasets like CIFAR and ImageNet?
Can this method be extended to domains beyond vision, such as NLP, to enhance adversarial robustness in text processing tasks?
What is the efficiency of the method, i.e, can it be implemented in plain GPUs?

---

### Official Review · Reviewer_J9Ns · 2023-10-26

**Soundness:** 2 fair
**Presentation:** 3 good
**Contribution:** 2 fair
**Rating:** 3
**Confidence:** 5

**Summary:**

The paper proposes SpecFormer, a novel approach to enhance ViTs’ resilience against adversarial attacks. It first provides a theoretical analysis that connects the local Lipschitz continuity of ViTs to their robustness against adversarial attacks. Based on this analysis, SpecFormer introduces a Maximum Singular Value Penalization (MSVP) algorithm that restricts the Lipschitz constant of the self-attention layers by penalizing the maximum singular values of the projection matrices. This helps regulate the extent of output variations when perturbations are introduced, making the model more robust. The authors integrates MSVP into ViT's training using the power iteration method to compute the maximal singular value of each attention layer. The authors then demonstrate the effevtiness of there approach on CIFAR, ImageNet and Imagenette datasets.

**Strengths:**

Given the current popularity of the self-attention layer, investigating the robustness of this type of layer is important. The current approach studies the local Lipschitz of the self-attention layer, which is a very interesting direction of research.

**Weaknesses:**

**In its current form, the paper has several weaknesses:**

**Major weaknesses:**
- The proposed bound takes advantage of the global Liscphitz of the linear mapping operations in the self-attention layer. This results in a very loose bound that is quadratic in N, the Frobenius norm of the input and delta. The looseness of the bound is clear when $\delta_0 = 0$.
- The proposed regularization based on the bound is a simple spectral regularization. However, it is not clear how the bound is affected since the regularization only focuses on the maximum singular values of the weights. The norm of the inputs may actually grow across the different layers of the model to compensate for the decrease of the spectral norm of the weights.
- I believe that the resulting robustness is a simple case of obfuscated gradients as described in [1]: the model is not particularly more robust, it is the attacks that are less effective, especially with such weak attacks as FGSM or PGD-2 with 2/255 of the attack budget. This misunderstanding is clear from these two statements:

``The proposition above suggests that when the maximum singular value of the weight matrices is small, the function fθ becomes less sensitive to input perturbations, which can significantly enhance adversarial robustness.``

``we can independently control the maximum singular values of these three weight matrices to imbue the model with adversarial robustness.``

The robustness from Lipschitz continuity is an inherent trade-off between the Lipschitz _and the margin_ of the classifier, as explicitly described in [2]. Reducing only the Lipschitz can have no impact on robustness if the margin is also reduced (which is often the case).

**On the choice of the norm:**
- The authors state: "The 2-norm is the most commonly used norm in Euclidean space". While the choice of the $\ell_2$ norm is perfectly reasonable, the $\ell_\infty$ norm is also highly studied in the adversarial robustness community. I would not say that the $\ell_2$ norm is "the most widely used norm".
- The authors state: "the sensitivity of a model to input perturbations is closely related to the 2-norm of the weight matrices". The sensitivity with respect to the $\ell_2$ norm is related to the 2-norm of the weight matrices, but one can still define the sensitivity with respect to any $\ell_p$ norm.
- The bound proposed in the paper is a bound on the local Lipschitz constant with respect to the $\ell_2$ norm. Therefore, it is not clear why the authors evaluate their approach against FGSM, which is a $\ell_\infty$ attack. It is also not clear whether the PGD attack used for the experiments is with respect to the $\ell_2$ or $\ell_\infty$ norm. A study of robustness against the linf norm can be done even though the proposed approach works for $\ell_2$, but mixing both $\ell_2$ and linf in the tables is confusing.

**On the structure of the paper:**
- In my opinion, the half page on page 4 on Lipschitz continuity could be condensed. Global Lipschitz and local Lipschitz are well-defined concepts in adversarial robustness.
- Theorem 4.2 is trivial and in my opinion not necessary.
- The whole section on power iteration is not necessary, the power method has been used many times in deep learning [2, 3, 4, 5, 6] and is a well-known concept. Also, the proof of convergence of the power iteration in the appendix doesn't seem to serve any clear purpose other than to fill space in the appendix. A reference to the relevant papers would suffice.
- Based on these comments, the authors could save space in the main paper and move Appendix D "Comparison with existing bounds" and the results on ImageNet to the main paper. Appendix D is interesting and would be better placed in the main paper.

[1] Athalye et al. Obfuscated Gradients Give a False Sense of Security: Circumventing Defenses to Adversarial Examples
[2] Tsuzuku et al. Lipschitz-Margin Training: Scalable Certification of Perturbation Invariance for Deep Neural Networks
[3] Farnia et al. Generalizable Adversarial Training via Spectral Normalization
[4] Meunier et al. A Dynamical System Perspective for Lipschitz Neural Networks
[5] Miyato et al. Spectral Normalization for Generative Adversarial Networks
[6] Yoshida et al. Spectral Norm Regularization for Improving the Generalizability of Deep Learning

**Questions:**

- Can the authors comment on the impact of the regularization on the bound, specifically with respect to the input norm of each layer?
- Can the authors about the norm used in the experimental settings?

---

### Official Review · Reviewer_Q4D5 · 2023-11-07

**Soundness:** 3 good
**Presentation:** 3 good
**Contribution:** 2 fair
**Rating:** 5
**Confidence:** 5

**Summary:**

The paper aims to improve the robustness of Vision Transformer using spectral normalization. During training the spectral norm of weight matrices in Transformer are estimated using power method and is added to the training loss. The theoretical results include a local Lipschitz bounds of self-attention which are correlated with the spectral norm of weight matrices.

The proof of this paper consists of two parts. The first part derives the Jacobian of one self-attention layer in Transformer. The second part bound the norm of the Jacobian around a  ball of input X. Although the theorem is presented and useful, I cannot regard this paper as a “theory” paper because the techniques used to derive and bound Jacobian are standard (e.g., using triangle inequalities). This is fine for a paper focusing more on the empirical side.

My main concern for this paper is that the theoretical results are loose and are not directly connected to the experiments part (see below); the empirical results used weak adversarial attacks and training methods that are far from the current state-of-the-art. So I tend to reject the current version of this paper, but I am happy to reevaluate the paper based on new results from the authors.

**Strengths:**

1. The robustness and Lipschitzness of transformer-based architectures are not well explored in the literature. A majority of papers studying robustness used convolutional neural networks. This paper nicely provides new results to fill this gap.

2. The theoretical results, while straightforward, are new in the sense that existing literature considers only the global Lipschitz bounds.

3. The writing of this paper is clear and easy to follow. Sufficient implementation details are given in the experiment section and also in the appendix.

**Weaknesses:**

1. Although the paper provides some bounds on local Lipschitz constants, it is rather loose and is likely vacuous in practical networks. In particular, the N(N+1)B factor can be really large. Here N is the number of tokens and B is the Frobenius norm of the input of this layer. The three factors multiplied together give a factor of about a thousand or more.

2. Some “theorems” and “propositions” (including 4.2 and 5.1) are textbook results. I feel it is not needed to write them as theorems in the paper. They do not add to the technical contribution of this paper.

3. The theory part and the experiments are disconnected: the main theorem presented is about the L2 norm of the attention layer, but the actual attack is done with PGD attack with Linf perturbation of 2/255 and 8/255.

4. The baselines presented in experiments are far from state-of-the-art. It would be better to actually show whether this method can improve upon a state-of-the-art adversarial defense training method. For example, https://robustbench.github.io lists many training methods can can achieve over 60% robust accuracy on CIFAR10.

**Questions:**

1. Can you evaluate the numerical values of the Lipschitz constants according to Theorem 4.3 for the networks trained in this work vs vanilla neural networks?

2. Can you provide results using PGD with more steps and also autoAttack? FGSM and 20 step PGD are pretty weak and cannot truly evaluate model robustness.

3. Can you provide results for L2 norm robustness? This is a more relevant setting to the theory developed in this work.